# Exploring the Association between Alcohol Drinking and Physical Activity in Adolescence; Two-Year Prospective Study in Younger Adolescents from Bosnia and Herzegovina

**DOI:** 10.3390/ijerph182211899

**Published:** 2021-11-12

**Authors:** Natasa Zenic, Małgorzata Lipowska, Dora Maric, Sime Versic, Hrvoje Vlahovic, Barbara Gilic

**Affiliations:** 1Faculty of Kinesiology, University of Split, Teslina 6, 21000 Split, Croatia; natasa@kifst.hr (N.Z.); sime.versic@kifst.hr (S.V.); 2Faculty of Social Sciences, Institute of Psychology, University of Gdańsk, 80-309 Gdańsk, Poland; malgorzata.lipowska@ug.edu.pl; 3PhD Program in Health Promotion and Cognitive Sciences, Sport and Exercise Research Unit, Department of Psychological, Pedagogical and Education Sciences, University of Palermo, 90144 Palermo, Italy; dora.maric@unipa.it; 4Faculty of Health Studies, University of Rijeka, 51000 Rijeka, Croatia; hrvoje.vlahovic@uniri.hr; 5Faculty of Kinesiology, University of Zagreb, 10000 Zagreb, Croatia

**Keywords:** physical activity, puberty, AUDIT, alcohol, substance misuse, sport

## Abstract

Insufficient physical activity and alcohol consumption (AC) are important health-threatening behaviors in adolescence, but there are controversial findings regarding the association that may exist between AC and physical activity levels (PALs) at this age. This study aimed to prospectively examine the relationship that may exist between AC and PAL in younger adolescence, considering the potential confounding effect of sports participation. The participants (*n* = 669, 337 females) were adolescents from Bosnia and Herzegovina who were tested on two occasions, at baseline (14 years of age) and again at follow-up (16 years of age). The variables included AC (as indicated by the AUDIT questionnaire), sports participation, age, gender (predictors), and PAL (obtained by the PAQ-A questionnaire) criterion. The results indicated that PAL at baseline was higher in those adolescents (boys) with a higher AUDIT score, but this association was partially confounded by sports participation. Multinomial regression indicated a higher risk of a decline in PAL over the study course in adolescents with a higher AUDIT score at baseline (OR = 1.32, 95%CI: 1.11–1.54 for being in the high-risk group for a decline in PAL). The most probable explanation is likely found in the high drop-out from sports in the studied period and earlier initiation of AC in adolescents involved in sports. Public health and sports authorities should urgently act preventively and develop educational programs against alcohol drinking in youth athletes.

## 1. Introduction

Adolescents are defined as individuals in the 10–19-year age group who are in a sensible life period characterized by significant physical, psychological, and social changes [1]. Moreover, adolescents form their habits and lifestyle choices during this period, which determine their lifestyle during adulthood. Unfortunately, the documented high prevalence of adolescents involved in behaviors that are detrimental to their health is highly alarming [2,3,4]. Among the most prevalent health-threatening behaviors are lack of physical activity and abuse of psychoactive substances, including the consumption of alcohol.

A sufficient physical activity level (PAL) has many health benefits, especially during adolescence. Precisely, sufficient PAL develops cardiovascular health, attains peak bone mass, optimizes muscular development, maintains a healthy weight, and prevents various chronic non-communicable diseases [5]. However, an alarmingly high rate (81%) of adolescents who are not sufficiently active is reported on a global basis [6]. This means that most adolescents do not reach the rate of 60 min of physical activity a day, as recommended by the World Health Organization [6]. Knowing that a sufficient PAL leads to many positive health outcomes and reduces the incidence of various diseases [5], a lack of PAL is considered a health-threatening phenomenon and has been investigated in detail during the last few decades [7,8]. Thus, as there has been global increased research interest in observing changes in PALs and their correlates, several studies have also been conducted in the territory of southeastern Europe. Specifically, prospective studies conducted on adolescents from Croatia and Bosnia and Herzegovina have reported a significant decline in PALs of 9% over a two-year period (from 16 to 18 years of age) [9,10] which is alarming, as the studied adolescents had insufficient PALs at baseline.

Another common health-threatening behavior in adolescence is the consumption of alcohol [11]. Drinking alcohol is considered a social behavior, meaning that adolescents under the great influence of their friends, parents, and family members start drinking even at a younger age, and afterward, consumption increases with friends and peers [12]. Increased alcohol consumption with advanced age occurs because drinking alcohol is used for socializing with friends and is considered socially desirable [13]. Thus, there is a high prevalence of adolescents that drink alcohol in many European countries [14], while the countries of southeastern Europe have recorded an alarmingly high incidence of harmful alcohol drinkers. Specifically, 29% of adolescents aged 16–18 years from Croatia [15], as well as 41% of boys and 37% of girls aged 17–19 years from Kosovo, have been shown to be harmful alcohol drinkers [16]. Similar figures were recorded in a study conducted on adolescents aged 17–18 years from Bosnia and Herzegovina (B&H), with 41% of boys and 27% of girls reported as being harmful alcohol drinkers [17], which places B&H among those European countries with the highest prevalence of alcohol drinking [18]. This is most likely the result of the culture itself, as drinking alcohol in some parts of B&H (depending on the religious affiliation of the inhabitants) is a socially acceptable and even preferable behavior [17]. The high prevalence of adolescents that drink alcohol should be taken seriously, as drinking alcohol leads to the deterioration of the cognitive functions of the brain, which is still in development, representing a significant threat to adolescents’ health [19]. 

As insufficient PAL and harmful alcohol drinking are both health-threatening behaviors that frequently occur in adolescents, several studies investigated the associations between PAL and alcohol drinking [16,20,21]. What is somewhat surprising is that studies have frequently reported a positive relationship between PAL and drinking alcohol, meaning that more active individuals also drink more alcohol [20,21]. This phenomenon has been explained by two facts. In brief, (first) a significant proportion of the PAL in adolescence is related to sports participation, while (second) alcohol is a major part of the gatherings after sports or exercise [20]. Thus, it has been continuously reported that adolescents involved in sports (i.e., more physically active) are more prone to drink alcohol than their non-athletic peers [15,22,23].

Collectively, the high prevalence of insufficiently active adolescents and adolescents who drink alcohol represents a major public health problem. Even though several studies have investigated the association between PAL and alcohol consumption, those studies were mainly conducted on older adolescents (>16 years of age) [15]. Meanwhile, there is a lack of studies that have prospectively investigated the relationship between alcohol drinking and PAL in younger adolescents (14–16 years of age), which is particularly important, knowing that studies conducted with older adolescents in the region (16–18 years) have evidenced the necessity of investigations in younger age since the results clearly point to the fact that alcohol initiation evidently happens at an earlier age [15]. Therefore, the aim of this study was to prospectively investigate the association between (baseline) alcohol consumption, and PAL changes in early adolescence (14–16 years of age). We hypothesized that alcohol drinking would have a negative effect on changes in PAL that occur between the ages of 14 and 16 years. As the public health authorities aim to reduce health-risk behaviors (alcohol drinking) and increase health-protective behaviors (physical activity), the results collected in this study could be of great use for designing health-promoting interventions. 

## 2. Materials and Methods

### 2.1. Participants and Study Design

The participants in this prospective study were 669 adolescents (337 females) from Herzegovina Neretva, Western Herzegovina, and Herzeg Bosnian County/Canton 10 in B&H. The average age of the participants at the study baseline was 14.7 ± 0.5 years. The sampling was based on a multistage cluster sampling method and included random selection of one-fourth of high schools in the studied cantons and random selection of one-third of the first-year classes in the selected schools. We intentionally observed regions inhabited almost exclusively by native Croats (almost exclusively Roman Catholics). Namely, in other parts of the B&H, the Islamic religion is common among native Bosniaks, and this fact would certainly influence the responses to questions about alcohol drinking, while due to the anonymity of the testing (please see later for details), we could not collect data about ethnicity or religious affiliations.

For the purpose of this investigation, we used the multistage cluster sampling method, including (i) random selection of 1/4 of high schools in the observed cantons and (ii) random selection of 1/3 of the first-year classes of these schools. Calculation of the appropriate sample size on the basis of the population (2662 first-graders), previously reported a prevalence of appropriate PAL in adolescents from the country (25%) [24], type I/II error rate of 0.05, and statistical power of 80%, the necessary sample size was 308 participants. The testing was organized across two time points: At baseline (late October to early November 2017, at the beginning of the participant’s first year of high school) and at the follow-up (May and June 2019, when the participants were at the end of the second year of high school). The investigation was approved by the ethical board of the Faculty of Kinesiology, University of Split, Croatia (EBO: 2181-205-05-02-05-14-005). Parental written consent was obtained before the first testing wave. 

Although during the baseline testing, 721 participants were involved, the final analysis included only participants that performed testing in both waves (*n* = 669; 337 females). The questionnaires were strictly anonymous and were performed online through an Internet-based application via participants’ personal cell phones. Testing was carried out during school hours, and the investigators had several cell phones that were used for responding to the survey if necessary. The participants received instructions to choose confidential codes at both baseline and follow-up in order to track their answers correctly. They had the option to refuse to participate and leave some questions or the entire questionnaire unanswered. 

### 2.2. Variables

Data were collected through previously validated questionnaires that included (i) sociodemographic variables and sports participation, (ii) data on alcohol consumption, and (iii) estimation of PALs [10,15,24]. Sociodemographic factors, sports participation, and alcohol consumption were evaluated at baseline, while PAL was estimated in both testing waves (Figure 1)

As a methodological remark we must note that we intentionally did not include sociodemographic variables and AUDIT at follow-up testing because of the following reasons. First, the main aim of the study was to observe the influence of the alcohol consumption on PAL at baseline and follow-up, and PAL changes over the study period. Therefore, it was crucial to test the subjects effectively, especially at the follow-up testing, which was done at the end of the second year of high-school, and to avoid any possible bias in testing of the criterion (PAL). 

The sociodemographic factors included the participants’ age (in years), gender (male or female), and self-reported socioeconomic status (above average, average, below average). Current sports participation was observed on a “Yes–No” scale. Consumption of alcohol was estimated with the Alcohol Use Disorders Identification Test (AUDIT), which consisted of 10 questions to which participants responded with a score of 0 (minimum) to 4 (maximum), for a hypothetical range of 0–40. AUDIT consists of three scores including dependence score, alcohol related problems, and consumption score. For instance, consumption score is composed from answers to questions such as “how often do you have a drink containing alcohol?“ or “how many drinks containing alcohol do you have on a typical day when you are drinking?” [25,26,27,28].

The Physical Activity Questionnaire for Adolescents (PAQ-A) was used to evaluate PAL. To assess PAL, the adolescents filled in the online form of the Physical Activity Questionnaire for Adolescents (PAQ-A). The adolescents filled the PAQ-A at baseline and at follow-up. The PAQ-A is a self-administered questionnaire designed for adolescents aged from 14 to 19 years, which includes questions regarding PA during the last seven days [29]. The PAQ-A consists of nine items assessing the frequency of participating in different types of PA (i.e., PA during physical education classes, school recess, free play, and sports). The results of each item and the total score range from 1 to 5, representing low to high PAL, respectively [30]. In this study, we observed the crude results of PAL at baseline (PAL-BL) and PAL at follow-up (PAL-FU). Next, crude PAL was also observed as a binomial variable with two categories: Results lower than 2.73 were classified as insufficient PAL, and results higher than 2.73 were marked as sufficient/normal PAL, as previously suggested [30]. Furthermore, to quantify the changes in PAL-BL and PAL-FU, we calculated the crude numerical difference between these two values (PALΔ = PAL-BL–PAL-FU). Next, we calculated the relative changes in PAL between baseline and follow-up (in %) using the following calculation: PALΔ% = (PAL-BL–PAL-FU)/PALBL * 100. For the purpose of later statistical calculations, participants were ordered according to their PALΔ%, and then grouped into three groups (Group 1: <33rd percentile; Group 2: 34th–66th percentile; Group 3: >66th percentile). The participants from Group 3 (with the greatest relative decline in PAL) were considered the “high-risk group,” Group 2 was considered the “medium-risk group,” and Group 1 was considered the “low-risk group” with regard to a decrease in PAL over the study period. Such categorization allowed us to calculate the multinomial regression for PALΔ% as a criterion, as recently suggested [31].

### 2.3. Statistics

The Kolmogorov–Smirnov test was used to define the normality of the distributions for all variables. 

Descriptive statistics included the calculation of means and standard deviations (for parametric variables) and frequencies and percentages (for non-parametric data). 

An independent samples *t*-test (for parametric variables), Mann Whitney test (MW) and a chi-square test (χ^2^ test) (for nonparametric variables) were used to identify differences between groups based on sufficiency/insufficiency of PAL at baseline and follow-up. To determine changes in PAL during the time points, a dependent samples *t*-test was used.

Logistic regression (with odds ratio (OR) and 95% confidence interval (CI) reported) was used to identify the associations between predictors obtained at baseline (sociodemographic variables, sports participation, and alcohol drinking) with dichotomized PAL (at baseline and follow-up (sufficient vs. insufficient PAL). All predictors were: (i) univariately correlated to PAL at baseline and follow-up, and then (ii) simultaneously included in the logistic regression in order to control the possible confounding effects on the established relationships. The model fit was checked by the Hosmer–Lemeshow test (with a significant χ^2^ indicating an inappropriate model fit). 

To analyze the associations between predictors obtained at baseline and the multinomial criterion (PALΔ%; high risk, medium risk, low risk for decline in PAL), multinomial logistic regression was used. Again, in the first phase of the regression calculation, all predictors were independently correlated with PALΔ%, and later, all predictors were simultaneously correlated with PALΔ% in a multiple regression manner in order to control the relationships for the potential influence of covariates. 

Statistica ver. 13.5 (Tibco Inc., Palo Alto, CA, USA) was used for all calculations, with a significance level of *p* < 0.05 applied for all calculations.

## 3. Results

PAL decreased significantly (*p* < 0.05) over the study course in the total sample (from 2.29 ± 1.13 to 2.13 ± 1.06), among boys (from 2.50 ± 1.00 to 2.31 ± 0.97), and among girls (from 2.15 ± 1.07 to 1.90 ± 0.91). 

No significant differences in age were found between groups according to the sufficiency/insufficiency of PAL at baseline and at follow-up (t = 0.37 and 0.66, *p* > 0.05). Also, no significant differences in socio-economic status were found between groups based on PAL sufficiency/insufficiency (MW = 4.57, and 2.14, for baseline and follow-up, respectively; both *p* > 0.05). 

Figure 2 presents the descriptive statistics and differences in AUDIT scores between groups based on sufficiency/insufficiency of PAL at baseline and follow-up. At baseline, adolescents with a higher AUDIT score were more likely to be sufficiently physically active (Figure 2A). However, when calculations of differences were stratified by gender, the differences reached statistical significance only for boys (Figure 2B), and not for girls (Figure 2C).

Table 1 presents the frequencies and percentages of the nominal predictors, as well as χ^2^ differences according to PAL sufficiency/insufficiency at baseline and follow-up. In brief, boys were more likely to be sufficiently physical active than girls, both at baseline (χ^2^ = 8.7, *p* < 0.01) and at follow-up (χ^2^ = 8.7, *p* < 0.05). Sports participation was significantly associated with sufficient PAL at baseline and at follow-up (χ^2^ = 59.91 and 54.98, respectively, both *p* < 0.001). 

At the univariate level, male gender, sports participation, and AUDIT score were significantly associated with PAL at baseline. Specifically, a higher likelihood for achieving an appropriate PAL was found for boys (OR = 1.68, 95%CI: 1.20–2.34), adolescents who participated in sports (OR = 2.41, 95%CI: 2.11–2.73), and adolescents who had higher scores on the AUDIT scale (OR = 1.41, 95%CI: 1.20–1.61). The Hosmer–Lemeshow test confirmed the appropriateness of the model fit (χ^2^ = 1.24–4.57, *p* > 0.05). The significance of the univariate predictors was confirmed even in the multivariate logistic regression calculation (ORs = 1.44, 2.17, and 1.37 and 95%CIs: 1.02–2.09, 1.81–2.61, and 1.01–1.68 for male gender, sports participation, and AUDIT score, respectively) (Figure 3A). 

When logistic regressions were calculated for dichotomized PAL at follow-up, the significant univariate predictors were male gender (OR = 1.59, 95%CI: 1.12–2.26) and sports participation (OR = 2.07, 95%CI: 1.71–2.50), with appropriate model fit as indicated by the Hosmer–Lemeshow test (χ2 = 1.14 and 3.71 for male gender and sports participation, respectively, *p* > 0.05). Sports participation remained the only significant predictor in the multivariate logistic regression (OR = 2.03, 95%CI: 1.68–2.47) (Figure 3B). 

Table 2 presents the results of the multinomial regression for the criterion risk of a decrease in PAL, where participants were grouped into three groups based on PALΔ%. At the univariate level, a higher AUDIT score at baseline was a significant predictor of high risk of a decrease in PAL over the study course (OR = 1.32, 95%CI: 1.11–1.54), while a medium risk of a decrease in PAL was evidenced for adolescents who were involved in sports at baseline (OR = 1.36, 95%CI: 1.06–1.74). The univariate results were confirmed even at the multivariate level (OR = 1.25, 95%CI: 1.03–1.57 for AUDIT as a predictor for a high risk of a decrease in PAL; OR = 1.36, 95%CI: 1.04–1.75 for sports participation at baseline as a predictor for a medium risk of a decrease in PAL). 

## 4. Discussion

Apart from a relatively well-known decrease in PAL in the period of early adolescence, this study revealed several important findings. First, PAL at baseline was higher in those adolescents who consume alcohol, and this is particularly characteristic of boys. Second, the association between PAL and alcohol consumption in both genders is confounded by sports participation, meaning that participation in sports is clearly the most influential factor of a higher alcohol consumption in adolescents who reported a higher PAL. Third, alcohol consumption observed at baseline (beginning of high school) is negatively associated with the PAL changes that occurred over the study course. Therefore, our initial study hypothesis can be accepted. 

### 4.1. Alcohol and PAL

From the perspective of public health, both a higher PAL and avoidance of substance misuse (SUM; including alcohol drinking) are considered as positive health behaviors [32,33]. As a result, it would be expected that lower likelihood of SUM would be related to a higher PAL among adolescents. However, our results indicate higher alcohol consumption in adolescents who reported a higher PAL, which is particularly characteristic for boys. Although this may seem surprising at first glance, the results are generally in accordance with previous studies declaring similar results globally [34,35,36,37,38,39]. Supportively, the review by Lisha and Sussman highlighted higher alcohol consumption in subjects with a greater PAL [40]. However, in explaining the cause–effect relationship between PAL and alcohol, various interpretations are offered, and previous research provides a handful of possible explanations. In our study, the background is partially related to sports participation and, consequently, higher PAL among adolescents who participate in sports. For a more profound discussion, it is important to obtain an overview of some contemporary theories that actually explain the background of the associations between sports and (general) SUM in adolescence. 

In their excellent overview, Wichstrøm and Wichstrøm logically presented the protective and risk factors associated with sports participation with regard to SUM [41]. First, they identified age segregation, time occupation, orientation toward success, and adult supervision as factors that could act as preventive against SUM in adolescence. Indeed, all of these factors actually prevent adolescents from SUM initiation. However, it cannot be ignored that sports participation is associated with a certain risk of SUM, since sports are a “social activity.” It logically increases the likelihood of being part of different social groups, where SUM is a common behavior, which can potentially lead to SUM initiation [41]. Therefore, it is crucial to elaborate why a sports society is an environment that influences higher alcohol consumption in adolescents.

First, alcohol is an inseparable part of sports culture and is often advertised in the context of sports [38]. It is advertised at sporting events and on television during sporting event broadcasts, while conversely, the number of messages warning of the negative consequences of alcohol consumption is minimal [38,42]. As a consequence, an environment that supports and approves the consumption of alcohol in the context of sports and sporting events is created. 

Second (next), the correlation of alcohol and PAL can be attributed to the sociopsychological theory of self-categorization, which states that people accept and adopt the beliefs, behaviors, and norms of their peers, i.e., members of the same group [43]. Namely, more physically active adolescents (i.e., those involved in sports clubs) establish close social ties with teammates, and through this, they develop their identity. They often have a need to feel a sense of belonging to a team, which they achieve through participation in group activities. This includes socially oriented activities that are often related to travel to competitions, but also celebrations after successful competitions and, thus, alcohol consumption [37,44]. Such behavior is not surprising, as adolescence is characterized by curiosity and exploration of new unfamiliar things, while alcohol is often the first substance with which adolescents become acquainted, since it is very accessible to youth [45,46]. 

### 4.2. Alcohol and PAL Changes

What is in no doubt the most important finding of this study is that alcohol consumption at baseline is associated with a relative PAL decline over the study course. Specifically, adolescents who consumed alcohol at the beginning of high school (age around 14 years) had a significant decline in PALΔ% during the following two years. There could be several possible explanations for these results.

First, it must be highlighted that adolescents who consumed alcohol at study baseline were involved in sports activities (please see previous discussion for more details). Moreover, it is clear that sports participation determines a large part of the total PAL in adolescents, and it has been proven that adolescents who participate in sports have a higher PAL than those who do not participate in sports [47,48]. However, it is known that the age of 10–14 years is the period in which children are most involved in sports, and then sports participation rapidly begins to decline [49]. Namely, this is the period when it has to be decided whether an individual will continue engaging in sports or quit. The reason for this could be found in the fact that in most countries, including B&H, sports are mainly focused on competitive success and a very small number of individuals continue to participate recreationally [9]. Thus, the association of high alcohol consumption at the beginning of the study when adolescents were still participating in sports and the PAL decline at the end of the study could be caused by the drop-out from sports during the course of the study.

Second, it is possible that some adolescents who consumed alcohol at the beginning of the study actually distanced themselves from a healthy lifestyle by lowering their PAL. Namely, the appearance of clustering health-risk behaviors such as unhealthy eating, drinking alcohol, smoking, and physical inactivity is well known [50,51]. In a large sample of Brazilian adolescents, 83% accumulated two or more risk factors (smoking, alcohol abuse, and physical inactivity) [32]. This could result in the fact that adolescents who drink alcohol are more likely to lean toward other health-risk behaviors, including reducing their PAL. 

Another possible explanation could be found in the assumption that adolescents involved in sports at the age of 14 are early maturers, as they are more predisposed to sports in physical and physiological terms. Early maturing adolescents are those that mature physically earlier than their peers and are therefore more predisposed to competitive sports achievement earlier in life [52]. However, for the purpose of this discussion, it is important to note that the observed studies showed pronation of early maturers to various eating and SUM disorders [53]. This most likely occurs due to fast and early biological changes, a lack of time for developing the social and cognitive skills necessary to cope with these changes, and a lack of peer support during significant social and bodily changes [54]. Thus, it is possible that early maturing adolescents were involved in sports at study baseline, and were exposed to the influence of older adolescents, which led to increased possibilities/risks of becoming involved in health-risk behaviors (i.e., the desire to prove themselves and the possibility of purchasing or procuring alcohol). Indeed, adolescents with better physical predispositions participate in sports activities with an older group of athletes, who mostly become their role models, making them more susceptible to following their behavior, including drinking alcohol [55]. 

### 4.3. Limitations and Strengths

The main limitation of this study comes from the fact that participants self-reported all variables. Therefore, they could lean toward socially desirable answers if they felt uncertain about their anonymity. However, this study was conducted in a region where alcohol drinking is socially acceptable behavior (please see Methods for details), and online questioning was applied, which likely reduced the possibility of participants not responding honestly. Second, PAL was evidenced by a questionnaire and not objectively measured, which was necessary due to anonymity. In addition, sport participation was observed on relatively short scale (Yes–No). However, this was done intentionally in order to preserve the anonymity of the participants (i.e., more detailed surveying could be evidenced as “targeting of certain individuals”). Moreover, this study was carried out in a specific region, and knowing the differences in religious affiliations in the country, further studies are needed to evaluate the studied relationships in other ethnicities. In addition, alcohol consumption has been assessed only at the beginning of the study, and future studies should investigate the alcohol consumption at follow-up in order to follow the relationship with PAL and its changes over time. It will definitively allow for more precise and elaborated discussion on such a highly important problem. Additionally, future studies should investigate this problem in more detail in order to widen the knowledge of the relationship between alcohol and physical activity in the general population of adolescents and in different contexts.

This is one of the rare studies where the influence of alcohol drinking on PAL was prospectively investigated in younger adolescents over a relatively long period of time (two years). This allowed us to objectively speak about the causality of the relationship between the studied variables. Moreover, apart from the targeted variables, this study included important covariates of PAL (i.e., sports participation), which allowed us to specifically discuss the observed relationships between the predictors and criteria. Knowing the importance of PAL, as well as the alarming decrease in PAL in this life period, the results can be used in the development of preventive targeted public health campaigns against decreases in PAL during adolescence. 

## 5. Conclusions

Sports participation is observed as an important determinant of PAL in adolescence, and this study confirmed such considerations. However, youth athletes regularly consume alcohol more than their non-athletic peers, which consequently results in finding a higher PAL among adolescents who drink. Although this association is partially moderated by male gender, it is still concerning that the association between (higher) alcohol drinking and (higher) PAL exists in the period of younger adolescence. 

Our results suggest that those adolescents who consume alcohol, although being more physically active at the age of 14–15 years, significantly decrease their PAL over the following 2 years. Almost certainly, the background of such an influence should be found in the drop-out from sports participation, which consequently results in a decreased PAL among former athletes. Meanwhile, it is possible that other factors, such as clustering health-risk behaviors, also resulted in such findings. 

Collectively, this research revealed specific interrelationships between alcohol drinking in adolescence and PAL changes, but also highlighted the complex sociocultural context of participation in competitive sports in the region where the sample was drawn from. Namely, although being clearly beneficial with regard to PAL in young adolescence (14–15 years of age), sports participation at this age seems to be accompanied by other unhealthy habits (e.g., drinking alcohol). Therefore, the drop-out from sports, which regularly and naturally occurs in the following period, will not result only in a decreased PAL but also in the retention of the drinking habits in (former) athletes. We consider that this study is of great importance for addressing two major public-health problems, i.e., alcohol drinking and insufficient physical activity in adolescents and points out the importance of future studies for further investigating this problem. As a result, public health and sports authorities must urgently develop educational programs aimed at the prevention of alcohol drinking in youth athletes. In addition, this research could be of interest to other readers such as physical education teachers and parents for perceiving the importance of detecting and later controlling unwanted health-risk behaviors in young adolescents.

## Figures and Tables

**Figure 1 ijerph-18-11899-f001:**
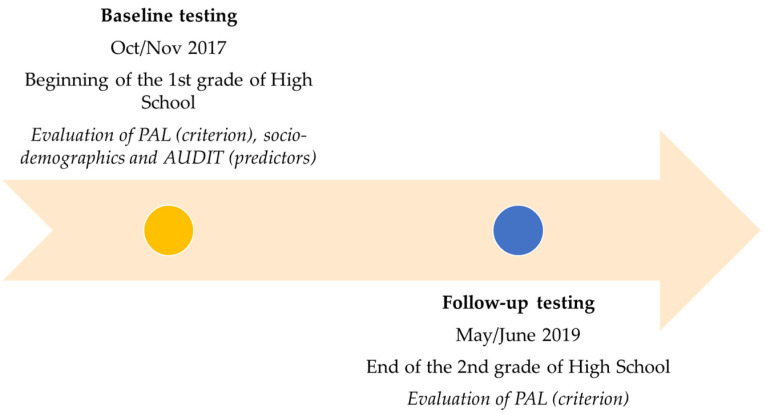
Testing scenario and variables observed at each testing wave.

**Figure 2 ijerph-18-11899-f002:**
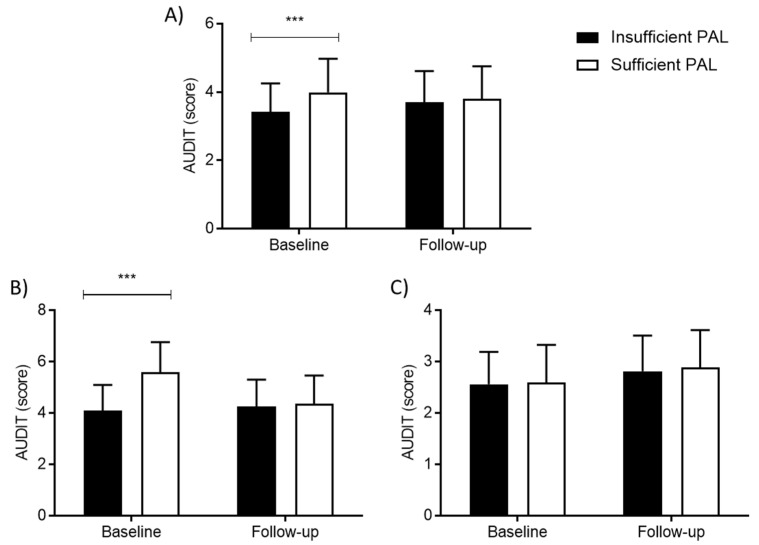
Descriptive statistics (means ± standard deviations) and *t*-test significance within groups for physical activity level for total sample (**A**), boys (**B**), and girls (**C**): *** 
*p* < 0.001).

**Figure 3 ijerph-18-11899-f003:**
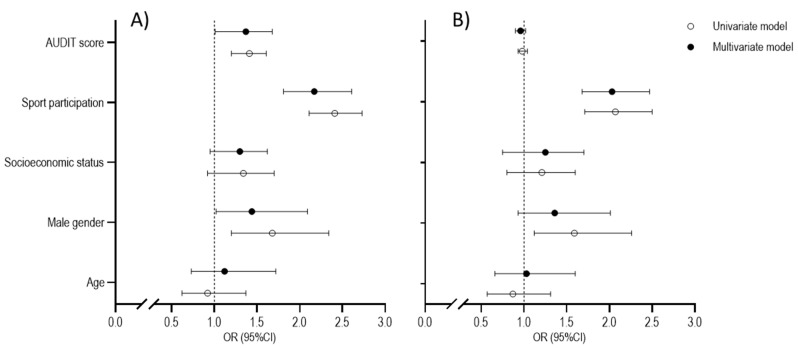
Results of the univariate and multivariate logistic regression calculations between predictors and criteria (PAL at baseline (**A**), and PAL at follow-up (**B**)).

**Table 1 ijerph-18-11899-t001:** Descriptive statistics (frequencies (F) and percentages (%)) and differences (Chi-square test-χ^2^) between groups based in physical activity level (PAL) sufficiency/insufficiency.

Variables	PAL Baseline			PAL Follow-Up		
	Insufficiency	Sufficiency	χ^2^	Insufficiency	Sufficiency	χ^2^
	F (%)	F (%)	(*p*)	F (%)	F (%)	(*p*)
Gender						
Boys	208 (62.65)	124 (37.35)	8.7	229 (69)	262 (77.7)	6.58
Girls	247 (73.3)	90 (26.7)	(0.01)	103 (31)	75 (22.3)	(0.02)
Sport participation						
No	389 (75.6)	125 (24.3)	59.97	413 (80.35)	101 (19.64)	54.98
Yes	66 (42.6)	89 (57.4)	(0.001)	78 (50.3)	77 (49.67)	(0.001)

**Table 2 ijerph-18-11899-t002:** Univariate and multivariate multinomial regression results for criterion “risk of PAL-decline over the study course” (low risk of PAL decline was set as the referent category).

	High Risk Group	Medium Risk Group	Low Risk Group
	OR (95% CI)	OR (95% CI)	OR (95% CI)
Univariate regression			
AUDIT score	1.32 (1.11–1.54)	1.01 (0.87–1.15)	REF
Male gender	1.05 (0.35–2.89)	0.49 (0.14–1.75)	REF
Socio-economic status	1.00 (0.51–1.54)	1.03 (0.61–1.45)	REF
Sport participation	1.16 (0.87–1.42)	1.36 (1.06–1.74)	REF
Age	1.16 (0.74–1.83)	1.02 (0.65–1.59)	REF
Multivariate regression			
AUDIT	1.25 (1.03–1.57)	1.00 (0.90–1.11)	REF
Male gender	1.30 (0.91–1.72)	1.01 (0.62–1.40)	
Socio-economic status	0.99 (0.41–1.60)	1.03 (0.56–1.56)	
Sport participation	1.08 (0.84–1.39)	1.36 (1.04–1.75)	
Age	1.14 (0.72–1.81)	0.98 (0.62–1.53)	

## Data Availability

Data will be provided to interested parties upon reasonable request.

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
