# Peer review of "Exploring the Association between Alcohol Drinking and Physical Activity in Adolescence; Two-Year Prospective Study in Younger Adolescents from Bosnia and Herzegovina"

_ijerph, 2021, doi:10.3390/ijerph182211899_

Round 1
Reviewer 1 Report
It is a very interesting paper. Revision is required for the following items.
First, the most important part is that there is not enough discussion about the variables that affect PAL. Since this study focuses on the effect of AC on PAL, variables that can affect PAL are not set as independent variables. Social and economic factors such as income and social relationship as well as individual income can affect PAL, but theoretical discussion and analysis on this have not been adequately conducted.
Second, the practical implication that this study can give is sufficient, but it is necessary to present what the theoretical implication is.
Third, the limitations of this study and the research topics necessary for the future should be presented.
Author Response
Please find responses in the enclosed file.

Reviewer 2 Report
The objective set out in the article is interesting and pertinent given the importance of establishing good habits that lead to a healthy lifestyle in adolescence, as these will last into adulthood. However, the study is carried out in a very specific context where, as the authors point out, various ethnic groups live together but have not been identified in the study, which would have been of great interest given the different relationship they have with alcohol consumption, as the authors also point out in the limitations.
On the other hand, reading the article reveals that there is a direct relationship with alcohol associated with post-competition celebrations as a result of competitive sport. I do not consider that this behaviour can be generalised in all adolescents or in all contexts.
From a methodological point of view, the article is well structured, the design is appropriate, as is the statistical treatment carried out. However, there is one aspect that I would like to be clarified: why was only alcohol consumption assessed at the beginning of the study? I think it would have been interesting to also assess alcohol consumption at the end of the study as was done for physical activity. This would have provided data on the evolution of alcohol consumption and its relationship with physical activity and cessation of physical activity.
In general, and from a methodological point of view, I consider that the article is good and I recommend its publication.

Author Response

(The authors gave the same response as above.)

Reviewer 3 Report
Dear authors,
The paper requires some improvements. The study explore the association between alcohol drinking and physical activity in adolescence (teenagers having 14 to 16 years old). In order to understand better these two type of issues is necessary to explain what is the meaning of alcohol consumption (any quantity, consuming alcohol every day, a certain volume of alcohol per day or per week?)? In the same time what suppose to be physical activity levels? Physical activity a certain number of hours per day, a total number of hours spent in organized physical activity per week, some extra hours based on voluntary effort? Are included teenagers coming from high schools with sport profile?
Then, what is the novelty of the paper and how the paper findings could support the domain? Can the results be considered useful to potential readers or researchers in the field? To what extension? Why is important such a research?
Maybe is necessary to explain who designed the questionnaire and based on what ? How the variables were established in the proposed research?
Because I am not sure that I understood very well, did the participants complete the same questionnaire after a period of 2 years or did there be changes in the questionnaire? If so, what and why?
I noticed that you declared for the questionnaire that you used for Current sports participation a “Yes–No” scale. This could be considered as sufficient?
In the same time you nee to reconsider to renumber parts from Figure 1 (as 1A, 1B and 1C, page 5) and from Figure 2 (as 2A and 2B, page 6).
One last thing. I think is important to check from the school curricula the number of compulsory hours of physical activity for each class of the schools. If there is a decreasing (starting with the first-year till the last year of the school), for sure this is not a good sign for your findings!
Author Response

(The authors gave the same response as above.)

Round 2
Reviewer 1 Report
All of things were well revised
Author Response
Thank you for recognizing the quality of our manuscript.
Reviewer 3 Report
Dear Authors,
I have few comments:
- In the Limitations section basically is fair to recognize that in the follow-up session you didn't investigate the alcohol consumption, but the paper objective is not to explore the association between alcohol drinking and physical activity? Here I am confused! You abdicated to check again this issue, during the follow-up session!
- If you didn't investigate the alcohol consumption, you are sure that your findings are relevant? You just suppose to be in the line with the findings related to the baseline testing!? Here I am very cautious, because I can tell you that when you signal to a young person something inappropriate related to his/her behavior, at least some of them respond positively (it is possible that some of the researched subjects have reviewed their behavior !). This could represent a weak point for your findings!
- I pointed out that you need to renumber some parts of your figures. I could tell that, this is not your strong point because you still have two figures numbered as Figure 2 (page 6 - which is OK, and page 7 which is not Ok). You should check also in the text how you used these numbers!
To be honest, I am not happy with your research itself! Is not so useful and not so well founded in terms of your approach, because you did not apply similar interview conditions !It went somewhat speculatively and not scientifically! There were questions with simple answers (such as Yes/No) that do not bring useful information and added value in research!
Author Response
I have few comments:
- In the Limitations section basically is fair to recognize that in the follow-up session you didn't investigate the alcohol consumption, but the paper objective is not to explore the association between alcohol drinking and physical activity? Here I am confused! You abdicated to check again this issue, during the follow-up session!
RESPONSE: Yes, indeed, we didn’t cover AUDIT across both measurements. Actually, in this study we were specifically focused on PAL, and the influence of initial alcohol consumption on PAL changes over the observed period. We absolutely agree that the information on alcohol consumption at both measurements would be beneficial, but there are two most important reasons why we didn’t observe it at the end of the study. First, the PAQ-A questionnaire is relatively long, and participants should be focused and concentrated. If AUDIT would be included as well, it will certainly detract participants which chill consequently compromise testing of the PAQ-A as the most important variable (outcome). Second, and even more important; the alcohol drinking habits (and AUDIT results) at the beginning of the high school were low. Meanwhile, it is relatively well known that during the first two years of high school adolescents initiate with (serious) alcohol drinking. As a result, if we asked participants about alcohol consumption on follow-up (end of the 2nd year of high school) we will certainly risk skipping of the questions and/or false-reporting.
We tried to explained it more specifically in this version of the manuscript and text reads: “Logistic regression (with odds ratio (OR) and 95% confidence interval (CI) re-ported) was used to identify the associations between predictors obtained at baseline (sociodemographic variables, sports participation, and alcohol drinking) with criteria-dichotomized PAL (sufficient vs. insufficient PAL) at baseline and follow-up (sufficient vs. insufficient PAL). All predictors were: (i) first univariately correlated to PAL at baseline and follow-up, and then the second phase were(ii) simultaneously included in the logistic regression in order to control the possible confounding effects on the established relationships.” (please see highlighted text in statistics subsection)
Also, for better understanding we amended the manuscript aim slightly, and it reads: “Therefore, the aim of this study was to prospectively investigate the association be-tween (baseline) alcohol consumption, and PAL changes in early adolescence (14–16 years of age).” (please see last paragraph of the Introduction)
- If you didn't investigate the alcohol consumption, you are sure that your findings are relevant? You just suppose to be in the line with the findings related to the baseline testing!? Here I am very cautious, because I can tell you that when you signal to a young person something inappropriate related to his/her behavior, at least some of them respond positively (it is possible that some of the researched subjects have reviewed their behavior !). This could represent a weak point for your findings!
RESPONSE: Yes, we must say that we agree with your observation about “responding positively”. However, we must note that in our case (our study) results on AUDIT scale were relatively low (logically, since we observed adolescents of relatively young age). Supportively, none of them reported “harmful alcohol drinking” (i.e. score of 11 and more; for details please see first paragraph of the Results where AUDIT scores are presented for boys and girls). Therefore, there was actually “no space for improvement”. As a result, we believe that skipping AUDIT at follow-up did not largely influence our results, especially considering that we weren’t focused on “changes in alcohol consumption over the study period” (this wasn’t the aim of the study). However, we tried to explain our point of view more specifically in this version of the paper, and text reads: “As a methodological remark we must note that we intentionally didn’t include sociodemographic variables and AUDIT at follow-up testing because of the following reasons. First, the main aim of the study was to observe influence of the alcohol consumption on PAL at baseline and follow-up, and PAL changes over the study period. Therefore, it was crucial to test the subjects effectively, especially at the follow-up testing which was done at the end of the 2nd year of high-school, and to avoid any possible bias in testing of the criterion (PAL).” (please see highlighted text after Figure 1 – testing scenario
- I pointed out that you need to renumber some parts of your figures. I could tell that, this is not your strong point because you still have two figures numbered as Figure 2 (page 6 - which is OK, and page 7 which is not Ok). You should check also in the text how you used these numbers!
RESPONSE: Please accept our apology for this mistake. The inclusion of the Figure 1 (new one) was the main reason for such nonconsistency. It is corrected now. Thank you.
To be honest, I am not happy with your research itself! Is not so useful and not so well founded in terms of your approach, because you did not apply similar interview conditions ! It went somewhat speculatively and not scientifically! There were questions with simple answers (such as Yes/No) that do not bring useful information and added value in research!
RESPONSE: Thank you one again for your comments and observations. We truly hope that the explanations we provided at least partially justified our decisions and eventual lacks in our research. With regard to “Yes/No” questions it was also done intentionaly, since it allowed to upkeep the anonymity of the subjects (i.e. more precise questioning on sport participation will almost certainly compromise the anonymity since it will allow “targeting” of certain participants who were involved in some kind of sport, and/or achieved sport competitive results). We can only promise that in future studies we will pay attention on your observations and consider all your suggestions in the forthcoming researches. In this version we additionally highlighted the mentioned problems and text reads:
“Also, sport participation was observed on relatively short scale (Yes – No). However, this was done intentionally in order to preserve the anonymity of the participants (i.e. more detailed surveying could be evidenced as “targeting of certain individuals”).”
“Also, alcohol consumption has been assessed only at the beginning of the study, and future studies should investigate the alcohol consumption at follow-up in order to follow the relationship with PAL and it’s changes over time. It will definitively allow more precise and elaborated discussion on such highly important problem.”
Thank you once again!